# FEDERATED CLASS-INCREMENTAL LEARNING: A HYBRID APPROACH USING LATENT EXEMPLARS AND DATA-FREE TECHNIQUES TO ADDRESS LOCAL AND GLOBAL FORGETTING

**Milad Khademi Nori**[*]
Department of Computer Science
Toronto Metropolitan University
Toronto, Ontario, Canada
mkn@torontomu.ca

**Il-Min Kim**
Electrical and Computer Engineering
Queen's University
Kingston, Ontario, Canada
ilmin.kim@queensu.ca

**Guanghui Wang**
Department of Computer Science
Toronto Metropolitan University
Toronto, Ontario, Canada
wangcs@torontomu.ca

## ABSTRACT

Federated Class-Incremental Learning (FCIL) refers to a scenario where a dynamically changing number of clients collaboratively learn an ever-increasing number of incoming tasks. FCIL is known to suffer from local forgetting due to class imbalance at each client and global forgetting due to class imbalance across clients. We develop a mathematical framework for FCIL that formulates local and global forgetting. Then, we propose an approach called Hybrid Rehearsal (HR), which utilizes latent exemplars and data-free techniques to address local and global forgetting, respectively. HR employs a customized autoencoder designed for both data classification and the generation of synthetic data. To determine the embeddings of new tasks for all clients in the latent space of the encoder, the server uses the Lennard-Jones Potential formulations. Meanwhile, at the clients, the decoder decodes the stored low-dimensional latent space exemplars back to the high-dimensional input space, used to address local forgetting. To overcome global forgetting, the decoder generates synthetic data. Furthermore, our mathematical framework proves that our proposed approach HR can, in principle, tackle the two local and global forgetting challenges. In practice, extensive experiments demonstrate that while preserving privacy, our proposed approach outperforms the state-of-the-art baselines on multiple FCIL benchmarks with low compute and memory footprints.

## 1 INTRODUCTION

Federated Learning (FL) is a distributed machine learning solution that enables clients to collaboratively optimize models without compromising their data privacy (Konečný et al., 2016). By exchanging model weights or gradients rather than centrally aggregating data, FL enhances data privacy while benefiting from the diversity of data to boost model robustness (McMahan et al., 2017). However, as the field of FL advances, new challenges arise, particularly in scenarios where the number of classes to be learned increases over time (Yoon et al., 2021). This scenario, known as Federated Class-Incremental Learning (FCIL), presents two issues because of the dynamic nature of incoming classes and the inherent class imbalance at both local (client-level) and global (system-wide) scales (Dong et al., 2022):

---

[*]MKN's personal email is miladkhademinori@gmail.com.

One of the two issues in FCIL is *local forgetting* (Dong et al., 2022), which occurs due to class imbalance within each client's dataset. This class imbalance arises because in FCIL, at a given time, clients possess data for only a subset of the classes, leading to biased learning and forgetting of previously learned classes. Concurrently, *global forgetting* (Dong et al., 2022), the second issue, arises because of class imbalance across different clients.

Furthermore, any proposed approach for FCIL not only needs to address the two aforementioned issues but also account for the hardware limitations (memory) often faced by clients. Specifically, it must cope with the incoming flow of tasks. These requirements set FCIL apart from the traditional federated learning scenario, underscoring the demand for tailored approaches (Babakniya et al., 2023; 2024).

The current approaches for FCIL fall into two categories: data-based, also known as exemplar-based (Dong et al., 2022) and data-free (Babakniya et al., 2023; 2024). Data-based approaches involve storing several exemplars for each class in the memory so that when the clients train their models on a new set of classes, they can use previous samples and form a mini-batch that statistically represents all the previous classes to mitigate forgetting. However, this approach may be impractical in FCIL due to memory constraints and presents privacy concerns. To remedy that, data-free approaches have recently gained popularity (Zhang et al., 2022; Gao et al., 2022; Shi & Ye, 2023; Zhang et al., 2023). These approaches usually rely on a generative model to produce synthetic data. However, data-free approaches may not be as performant as the data-based ones (Dong et al., 2022).

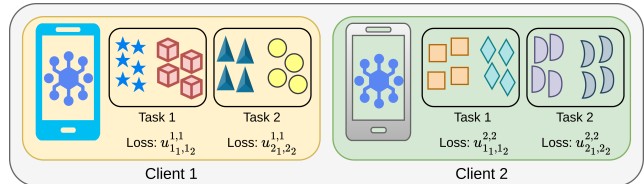

Figure 1: Local and global forgetting occur due to class imbalance at both the local (client-level) and global (system-wide) scales. Whereas local forgetting refers to the combination of sub-optimal losses for the previous tasks and intra-client task confusion, global forgetting involves inter-client task confusion.

To harness the advantages of both data-based and data-free approaches, we present Hybrid Replay (HR), a unified approach with low memory overhead and high performance. Our contributions in this work are as follows:

- We develop a mathematical framework for FCIL to prove that our proposed approach HR can, in principle, tackle the two local and global forgetting challenges.

- HR unifies two approaches : (i) HR incorporates our efficient version of exemplar replay based on our *customized autoencoder*. Our efficient exemplar replay inflicts an order of magnitude less memory overhead than vanilla exemplar-based approaches. (ii) HR includes our version of an autoencoder-based data-free approach with almost no memory overhead. Our data-based and data-free approaches are distinct from those in the literature, as we utilize a new strategy and a *novel autoencoder* to implement these approaches.

- The backbone of HR, as mentioned above, is our customized autoencoder that is used for both data classification and the generation of synthetic data. To determine the embeddings of new tasks for all clients in the latent space of the encoder, the server uses the Lennard-Jones Potential formulations (Jones, 1924). Meanwhile, at the clients, to overcome local forgetting, the decoder decodes the stored low-dimensional latent space exemplars back to the high-dimensional input space, used to address local forgetting. To overcome global forgetting, the decoder generates synthetic data in a data-free manner.

- Extensive experiments demonstrate that while preserving privacy, our proposed approach outperforms the state-of-the-art baselines on multiple FCIL benchmarks with low compute and memory footprints.

## 2 RELATED WORK

**Federated Learning** (FL) can collaboratively train *robust* models by utilizing the *diverse* local data of many clients while promising an enhanced level of privacy (Yurochkin et al., 2019; Wang et al., 2020; Li et al., 2021; Yang et al., 2021). In FL, the server aggregates and averages the weights of

the local models of different clients (McMahan et al., 2017). Follow-up works focus on mitigating the drift problem of weights at different clients (Shoham et al., 2019; Li et al., 2020), or aim to minimize the communication overhead (Chen et al., 2019; Zhang et al., 2020b;c). Fast convergence (Karimireddy et al., 2020) and personalization (Fallah et al., 2020; Lange et al., 2020) have also been studied. Unsupervised FL has shown promising results (Peng et al., 2019; Zhang et al., 2020d; Dong et al., 2021; Liu et al., 2021a; Chu et al., 2022; Liu et al., 2022). However, current FL approaches cannot work in scenarios where the number of classes grows constantly. This is because the clients have memory limitations and cannot accumulate all the previous classes (Hamer et al., 2020; Lyu & Chen, 2021).

**Class-Incremental Learning** (CIL) is a scenario where the model has to visit a stream of classes (Zhang et al., 2020a; Khademi Nori & Kim, 2025), learn them incrementally (Ahn et al., 2021), and discriminate them all without forgetting earlier classes (Kim & Choi, 2021). Regularization is one of the earliest attempts to mitigate catastrophic forgetting (Kirkpatrick et al., 2017). Knowledge distillation (Li & Hoiem, 2017) has been shown effective in retaining the knowledge of previous classes (Shmelkov et al., 2017). Generative models have been adopted to produce synthetic data to be interleaved with the new classes to mitigate biased learning and forgetting (Shin et al., 2017; Wu et al., 2018). It has been frequently noted that class imbalance is the most important obstacle in CIL (Douillard et al., 2020; Liu et al., 2020; Rebuffi et al., 2017; Wu et al., 2019). Bias-correction (Liu et al., 2021b; Yan et al., 2021) and knowledge distillation (Hu et al., 2021; Simon et al., 2021) have been shown effective in overcoming class imbalance. Hybrid replay approaches integrate model-based replay (generative modeling) and exemplar-based replay. In HR, instead of storing raw exemplars, latent space features are stored, which require orders of magnitude less memory (Hayes et al., 2020; Van de Ven et al., 2020; Wang et al., 2021; Tong et al., 2022; Zhou et al., 2022). Table 1 compares our proposed work HR and other hybrid baselines such as Remind (Hayes et al., 2020), Remind+ (Wang et al., 2021), and i-CTRL (Tong et al., 2022).

**Federated Class-Incremental Learning** (FCIL) is a scenario where multiple distributed clients *collaboratively and incrementally* learn new classes over time without sharing their data, facing both challenges of *local and global forgetting* due to imbalanced (non-IID) data distributions (Yoon et al., 2021; Dong et al., 2022). FCIL approaches fall into two categories of (i) exemplar-based approaches and (ii) data-free approaches. (i) As an exemplar of exemplar-based, Global-Local Forgetting Compensation (GLFC) model integrates class-aware gradient compensation and class-semantic relation distillation to mitigate local forgetting, while a proxy server and a prototype gradient-based communication mechanism tackle global forgetting (Dong et al., 2022).

(ii) Data-free approaches. Relation-guided Representation Learning for Data-Free Class-Incremental Learning (R-DFCIL) employs relational knowledge distillation to maintain compatibility between new and previous class representations (Gao et al., 2022). The Federated Fine-Tuning with Generators (FedFTG) employs knowledge distillation and hard sample mining to optimize the global model and address data heterogeneity across clients (Zhang et al., 2022). TARGET (federated class-continual learning via exemplar-free distillation) uses a pre-trained global model for knowledge transfer and a generator to simulate data distribution, without needing additional datasets or storing private data (Zhang et al., 2023). In (Babakniya et al., 2023), a server-trained generative model synthesizes past data samples from discriminators and then trains generators that are given to clients, enabling clients to locally mitigate catastrophic forgetting without compromising privacy.

The Local-Global Anti-forgetting (LGA) model uses category-balanced, gradient-adaptive compensation loss and semantic distillation to balance learning among local clients, while a proxy server manages global forgetting through self-supervised prototype augmentation (Dong et al., 2023). FedSpace adapts to asynchronous and varied task sequences of clients using prototype-based learning, representation loss, fractal pre-training, and a modified aggregation policy (Shenaj et al., 2023). Shi & Ye (2023) introduces a prototype reminiscence mechanism that dynamically reshapes old class feature distributions by integrating previous prototypes with new class features, in tandem with an augmented asymmetric knowledge aggregation. An improved version of (Babakniya et al., 2023) has been proposed by Babakniya et al. (2024).

Table 1: Comparison of our approach with other hybrid approaches Remind (Hayes et al., 2020), Remind+ (Wang et al., 2021), and i-CTRL (Tong et al., 2022).

| | Classification Approach | Quantization | Learning Scenario | Latent Space Representation | Architectural Simplicity |
|---|---|---|---|---|---|
| HR | Classification occurs within the latent space of the encoder, leveraging Euclidean distance. | Incorporating quantization within our architecture would be straightforward, further improving performance. | Offline learning. | Uses a structured latent space via the Lennard-Jones Potential formulations. | Our work showcases clean and minimalistic designs, as depicted in Fig. 2. |
| Remind | Classification occurs after the decoding process, employing a cross-entropy loss function as depicted in Fig. 2 of Remind. | Applies quantization to the latent exemplars to enhance compression. | Online learning. | Uses an unstructured latent space for its CNNs. | Fig. 2 of Remind shows an ad hoc and unnecessarily complex architecture, complicating implementation and scalability. |
| Remind+ | Classification occurs after decoding with a cross-entropy loss function. | Incorporates quantization for feature compression. | Online learning. | Relies on an unstructured latent space for its autoencoder. | Figs. 2 and 3 of Remind+ demonstrate ad hoc and overly complex designs, hindering scalability. |
| i-CTRL | Classification within the latent space of the encoder, leveraging Euclidean distance. | Incorporating quantization within their architecture would be straightforward, further improving performance. | Offline learning. | Uses a structured latent space via Linear Discriminative Representation. | Has a clean and minimalistic design, as depicted in Fig. 1 of i-CTRL. |

## 3  FCIL PROBLEM FORMULATION

In traditional machine learning settings (supervised learning), models are trained using a fixed dataset, where the class distribution is known in advance and the training data are accessible during the model's training phase. The objective is typically to minimize a loss function that measures the discrepancy between the predicted labels and the true labels:

$$I_{\boldsymbol{\theta}} = \int_{\mathcal{X} \times \mathcal{Y}} \ell(f_{\boldsymbol{\theta}}(\boldsymbol{x}), y) p(\boldsymbol{x}, y) \, d\boldsymbol{x} \, dy, \tag{1}$$

where $\mathcal{X}$ and $\mathcal{Y}$ represent the space of inputs and labels, respectively, $f_{\boldsymbol{\theta}}$ denotes the model parameterized by $\boldsymbol{\theta}$, $\ell$ is the loss function, and $p(\boldsymbol{x}, y)$ is the joint distribution of inputs and labels. When $\mathcal{Y}$, the label set, has $N$ classes, the loss function in Eq. 1 can be specifically written as:

$$I_{\boldsymbol{\theta}} = \frac{1}{N-1} \sum_{i=1}^{N} \sum_{j=i+1}^{N} u_{ij}, \quad \text{with} \quad u_{ij} = \int_{\mathcal{X} \times \{i,j\}} \ell(f_{\boldsymbol{\theta}}(\boldsymbol{x}), y) p(\boldsymbol{x}, y) \, d\boldsymbol{x} \, dy. \tag{2}$$

Eq. 2 reflects the need to evaluate the model's performance across all possible pairs of class labels, ensuring that each pair is considered in the loss calculation.

When we introduce the concept of tasks in CIL, $k$th task $T_k$ encompasses a specific subset of classes from $\mathcal{Y}$. The model must adapt its parameters to minimize the loss across successive tasks while maintaining robust performance on previous tasks. Each task $T_k$ involves learning a designated set of classes $\mathcal{Y}_k \subset \mathcal{Y}$. Assuming all tasks contain the same number of classes, denoted by $N$, the cumulative loss function across tasks can be modeled as:

$$I_{\boldsymbol{\theta}} = \sum_{k=1}^{K} \sum_{l=1}^{K} \frac{1}{N^2} \sum_{i_k=1}^{N} \sum_{j_l=1}^{N} u_{i_k, j_l}, \quad \text{with} \quad u_{i_k, j_l} = \int_{\mathcal{X} \times \{i_k, j_l\}} \ell(f_{\boldsymbol{\theta}}(\boldsymbol{x}), y) p(\boldsymbol{x}, y) \, d\boldsymbol{x} \, dy. \tag{3}$$

Here, $K$ represents the total number of tasks, and $N$ is the uniform number of classes per task.

For a new task $T_{k+1}$ that includes $N$ classes, the cumulative loss function after incorporating this task can be recursively updated from the previous tasks as follows:

$$I_{\boldsymbol{\theta}}^{(k+1)} = I_{\boldsymbol{\theta}}^{(k)} + \frac{1}{N^2} \left( \sum_{i=1}^{N} \sum_{j=1}^{N} u_{i_{k+1}, j_{k+1}} + 2 \sum_{l=1}^{k} \sum_{i=1}^{N} \sum_{j=1}^{N} u_{i_l, j_{k+1}} \right), \tag{4}$$

where $I_{\boldsymbol{\theta}}^{(k)}$ represents the cumulative loss function up to task $k$. The term $\sum_{i=1}^{N} \sum_{j=1}^{N} u_{i_{k+1}, j_{k+1}}$ accounts for the interactions within the new task $T_{k+1}$, and the term $\sum_{l=1}^{k} \sum_{i=1}^{N} \sum_{j=1}^{N} u_{i_l, j_{k+1}}$,

multiplied by 2, captures the bidirectional interactions between all classes of the new task and each class of the previously learned tasks. The normalization factor $\frac{1}{N^2}$ ensures uniform weighting of class interactions across all tasks.

In FCIL, the model evolution takes advantage of collaborative contributions from different clients, each maintaining its local dataset. As new tasks and their associated classes arrive at different clients, each client updates its model to accommodate new knowledge while ensuring coherence with the collective knowledge maintained across the federated network.

The recursive update formula for the global loss function, as formulated in Eq. 5, encapsulates this dynamic. Here, $I_{\boldsymbol{\theta}}^{(k+1)}$ represents the global loss after incorporating task $(k+1)$. This is computed by aggregating updates from each client pair $(m, n)$, accounting for both their individual and mutual contributions:

$$I_{\boldsymbol{\theta}}^{(k+1)} = \sum_{m=1}^{M} \sum_{n=1}^{M} \left( I_{\boldsymbol{\theta}}^{(k),m,n} + \Delta I_{\boldsymbol{\theta}}^{(k+1),m,n} \right), \tag{5}$$

where

$$\Delta I_{\boldsymbol{\theta}}^{(k+1),m,n} = \frac{1}{N^2} \left( \sum_{i=1}^{N} \sum_{j=1}^{N} u_{i_{k+1},j_{k+1}}^{m,n} + 2 \sum_{l=1}^{k} \sum_{i=1}^{N} \sum_{j=1}^{N} u_{i_l,j_{k+1}}^{m,n} \right). \tag{6}$$

Eqs. 5 and 6 outline the critical balance required in FCIL: the preservation of knowledge gained in the past, represented by $I_{\boldsymbol{\theta}}^{(k),m,n}$. This preservation is crucial for maintaining system stability (stability-plasticity dilemma) and is typically achieved through mechanisms such as regularization, knowledge distillation, and replay.

Meanwhile, in Eq. 6 there are two loss terms, the first loss term has to do with learning the new task and encompasses all the interactions between classes in the new task, and the second loss term, which is very important, accounts for the loss terms between the new task and all the previous tasks. Failing to minimize the second loss would result in *intra-client task confusion*, where the model fails to differentiate between tasks within the same client. There is also another task confusion implicit in Eq. 5 that we call *inter-client task confusion* addressing the loss term between different tasks at different clients. Failure to minimize inter-client task confusion will render the model incapable of distinguishing between different classes of different clients. Thus, overall, there are four challenges that need to be considered: (i) preserving prior knowledge, (ii) effectively integrating new tasks, (iii) mitigating intra-client task confusion, and (iv) resolving inter-client task confusion. In the literature (i) and (iii) together are called local

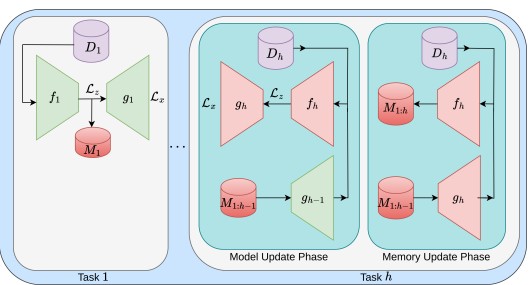

Figure 2: Our HR approach, except for Task 1, consists of two key phases for handling tasks in FCIL. For Task 1, the autoencoder is trained, and the compact latent representations $M_1$ are stored. For subsequent tasks $h$, in the *Model Update Phase*, compact representations from previous tasks $M_{1:h-1}$ are decoded, interleaved with the new task's data $D_h$, and used to update the model. In the *Memory Update Phase*, with the updated model, compact representations for both the old tasks and the new task are computed and stored $M_{1:h}$.

forgetting while (iv) is called global forgetting (Dong et al., 2022). In the next section, we will present our proposed approach and discuss how our proposed approach tackles these challenges.

## 4  PROPOSED APPROACH: HYBRID REPLAY

Our proposed approach (shown in Fig. 2), Hybrid Replay (HR), is based on a specialized autoencoder that serves two primary functions: data classification and synthetic data generation. This customized autoencoder comprises an encoder and a decoder operating as follows. The encoder $f(\boldsymbol{x}) : \mathbb{R}^n \to \mathbb{R}^m$ maps the input data $\boldsymbol{x} \in \mathbb{R}^n$ to a low-dimensional latent representation $\boldsymbol{z} \in \mathbb{R}^m$. The decoder $g(\boldsymbol{z}) : \mathbb{R}^m \to \mathbb{R}^n$ then reconstructs the input data from this latent representation.

While traditional Variational Autoencoders (VAE) (Kingma & Welling, 2013) focus primarily on generalization and generating new images, HR adapts the VAE framework to balance modeling data

distributions with enhanced class-specific clustering in the latent space. Accordingly, the modified loss function incorporates the standard VAE loss terms with an additional loss term designed to cluster samples around their class centroids:

$$
\boldsymbol{L}(\boldsymbol{x}, \hat{\boldsymbol{x}}, \boldsymbol{z}) = \text{ELBO}(\boldsymbol{x}, \hat{\boldsymbol{x}}, \boldsymbol{z}) + \lambda \boldsymbol{L}_{\boldsymbol{z}}(\boldsymbol{z}, \boldsymbol{p})
$$

$$
= -\mathbb{E}_{q(\boldsymbol{z}|\boldsymbol{x})}[\log p(\boldsymbol{x}|\boldsymbol{z})] + \text{KL}(q(\boldsymbol{z}|\boldsymbol{x})\|p(\boldsymbol{z})) + \lambda \sum_{i=1}^{K} \sum_{j=1}^{N} \|\boldsymbol{z}_{i_j} - \boldsymbol{p}_{i_j}\|^2 \qquad (7)
$$

where ELBO represents the Evidence Lower Bound, KL denotes the Kullback-Leibler divergence, $\|\cdot\|^2$ indicates the $L^2$ norm, $\boldsymbol{p}_{i_j}$ represents the centroid embedding for the $i$-th task and $j$-th class, and $\lambda$ is a hyperparameter.

To determine the embeddings of new task centroids for all clients in the encoder's latent space, the server utilizes the Lennard-Jones Potential formulations (Jones, 1924). Specifically, the total potential energy $\mathcal{U}$ of the system, which represents the interactions between all pairs of class centroid embeddings, is calculated as follows:

$$
\sum_{i,j=1}^{K,N} \sum_{i',j' \neq i,j} 4\varepsilon \left[ \left( \frac{\sigma}{\|\boldsymbol{p}_{i'_{j'}} - \boldsymbol{p}_{i_j}\|} \right)^{12} - \left( \frac{\sigma}{\|\boldsymbol{p}_{i'_{j'}} - \boldsymbol{p}_{i_j}\|} \right)^6 \right], \qquad (8)
$$

where $\|\boldsymbol{p}_{i'_{j'}} - \boldsymbol{p}_{i_j}\|$ is the Euclidean distance between the centroid embeddings. The parameters $\varepsilon$ and $\sigma$ determine the depth of the potential well and the equilibrium distance, respectively.

To update the centroid embeddings, the server iteratively adjusts each $\boldsymbol{p}_{i_j}$ using the gradient of the potential energy, optimized with a learning rate $\eta$. The update can be mathematically expressed as:

$$
\boldsymbol{p}_{i_j} \leftarrow \boldsymbol{p}_{i_j} - \eta \sum_{i',j' \neq i,j} 24\varepsilon \left[ 2 \left( \frac{\sigma^{12}}{\|\boldsymbol{p}_{i'_{j'}} - \boldsymbol{p}_{i_j}\|^{13}} \right) - \left( \frac{\sigma^6}{\|\boldsymbol{p}_{i'_{j'}} - \boldsymbol{p}_{i_j}\|^7} \right) \right] (\boldsymbol{p}_{i_j} - \boldsymbol{p}_{i'_{j'}}). \qquad (9)
$$

**Leonard-Jones formulations** laid out above determine the global alignment of centroids for new classes in new tasks. As described in lines 6–8 of Algorithm 1, where clients compute the unaligned embeddings for new classes and send them to the server. Then the server aligns these centroids and sends them back to the clients. Clients use these aligned centroids to train on new tasks, ensuring minimal overlap with existing classes globally.

**Memory update routine** at the clients follows the well-known fixed memory procedure (Rebuffi et al., 2017) (as opposed to growing memory design (Masana et al., 2020)) where the number of exemplars of the classes decreases as new classes arrive (Dong et al., 2022). To not cause clutter we refrain from reciting those well-known routines.

**Synthetic data generation for replay** operates through a dual-path mechanism, as outlined in Algorithm 2. For each client, if exemplars of a particular class from previous tasks exist in the client's memory $\mathcal{M}_{i_j}^c$, they are passed through the decoder $g(\boldsymbol{\theta}_{h-1})$ to regenerate the corresponding samples (Line 6 of Algorithm 2). This ensures faithful reproduction of previously encountered data, preventing local forgetting. If exemplars are absent, the centroid embedding $\boldsymbol{p}_{i_j}^c$ of that class is perturbed with Gaussian noise $\mathcal{N}(0, \sigma^2)$ (Line 8 of Algorithm 2) and then decoded to generate synthetic samples. This method enables the client to generate synthetic data even for tasks or classes it has not encountered directly, effectively preventing global forgetting.

---

**Algorithm 1** HR Algorithm for FCIL at the Server

1: **Inputs:** $R$: total clients, $K$: tasks, $I$: clients per round, $L$: local epochs, $\Omega$: communication rounds
2: **Output:** Optimized autoencoder $\boldsymbol{\theta}_K^*$
3: **Initialize:** $\boldsymbol{\theta}_0$ at the server
4: **for** $h = 1$ to $K$ **do**
5:     Select $I$ clients randomly from $R$
6:     Receive unaligned centroids $\{\boldsymbol{p}_{h_j}\}$ from clients for task $h$
7:     Align centroids $\{\boldsymbol{p}_{h_j}\}$ for task $h$ via Eq. 9
8:     Broadcast aligned centroids $\{\boldsymbol{p}_{h_j}\}$ for task $h$ to clients
9:     **for** $\omega = 1$ to $\Omega$ **do**
10:         **for** each client $i$ from selected clients **in parallel do**
11:             Update $\boldsymbol{\theta}_h$ at client $i$ using Algorithm 2
12:         **end for**
13:         Receive $\boldsymbol{\theta}_h^c$ from clients after communication round $\omega$
14:         $\boldsymbol{\theta}_h \leftarrow \frac{1}{I} \sum_{c=1}^{I} \boldsymbol{\theta}_h^c$   ▷ FedAvg (McMahan et al., 2017)
15:         Broadcast $\boldsymbol{\theta}_h$ to clients
16:     **end for**
17: **end for**

**Classification** in the model is performed by mapping input $\boldsymbol{x}$ through the encoder $f(\boldsymbol{\theta}^*, \boldsymbol{x})$ directly. The predicted class for $\boldsymbol{x}$ is determined by $\operatorname{argmin}_{c,i,j} \| f(\boldsymbol{\theta}^*, \boldsymbol{x}) - \boldsymbol{p}_{i_j}^c \|$ where $\| \cdot \|$ denotes the Euclidean distance. This metric identifies the class whose centroid $\boldsymbol{p}_{i_j}^c$ from client $c$ is nearest to the encoded representation of $\boldsymbol{x}$.

**Local and Global Forgetting** (Dong et al., 2022) has been detailed in the previous section in our mathematical framework where in Eqs. 5 and 6 we identified four primary challenges in FCIL: (i) preserving prior knowledge, (ii) effectively integrating new tasks, (iii) mitigating intra-client task confusion, and (iv) resolving inter-client task confusion. In the literature, the challenges of preserving prior knowledge and mitigating intra-client task confusion are collectively referred to as local forgetting, while resolving inter-client task confusion is termed global forgetting (Dong et al., 2022). In the following, we discuss how Algorithm 1 and Algorithm 2 overcome local and global forgetting:

During the learning of task $h$, local forgetting is addressed through a dual strategy: replaying previously seen data (Lines 7 and 9 of Algorithm 2) and applying knowledge distillation (Lines 19, 20, and 21 of Algorithm 2) to retain knowledge from earlier tasks. In contrast, global forgetting is handled differently. Losses related to task $h$ across different clients, as expressed in Eqs. 5 and 6 and shown in Fig. 1, cannot be minimized during the same session due to the unavailability of the relevant data or its synthetic representation at the other clients. Instead, this minimization is deferred to the $(h+1)$ session, where clients receive the class centroid embeddings $\{\boldsymbol{p}_{i_j}\}$ (Line 8 of Algorithm 1). These embeddings enable clients to generate synthetic data representing tasks from other clients, helping the optimization of cross-client task losses and mitigating global forgetting.

## 5 EXPERIMENTS

Following the simulation settings outlined by Dong et al. (2022), our experimental evaluations were conducted across five benchmarks: CIFAR-100 (10/10/50/5 and 20/5/50/5), ImageNet-Subset (10/20/100/10 and 20/10/100/10), and TinyImageNet (10/5/300/30), where $(A/B/C/D)$ indicates the simulation configuration, with $A$ denoting the number of tasks, $B$, classes per task, $C$, number of clients, and $D$, the number of active clients per round. The inclusion of both short and long task sequences for CIFAR-100 and ImageNet-Subset allows us to assess the robustness of different baselines across varying task lengths (Masana et al., 2020). We utilized a ResNet-18 architecture for the discriminator/encoder $f()$, as referenced in (Babakniya et al., 2023; 2024), and a four-layer CNN for the generator/decoder $g()$. The data for each task is distributed among clients using Latent Dirichlet Allocation (LDA) with the parameter $\alpha = 1$. Clients utilize an SGD optimizer for local model training. The reported results represent the averages from ten random initializations accompanied by SEMs.

---

**Algorithm 2** HR Algorithm for FCIL at Clients

---
1: **Input:** $\{T_1, T_2, \ldots, T_K\}$, $\{f(\boldsymbol{\theta}_0), g(\boldsymbol{\theta}_0)\}$, memory $\mathcal{M}_0$
2: **Output:** $\boldsymbol{\theta}_K^*$, memories $\mathcal{M}_K^*$, and $\{\boldsymbol{p}_{i_j}^c\}$
3: **for** $h = 1$ to $K$ **do**
4:      Initialize dataset $D_h \leftarrow T_h$
5:      **for** each client $c$, each previous task $i < h$, each class $j$ **do**
6:          **if** memory $\mathcal{M}_{i_j}^c$ exists **then**
7:              $D_h \leftarrow D_h \cup g(\boldsymbol{\theta}_{h-1}, \mathcal{M}_{i_j}^c)$   ▷ Use memory data to synthesize samples
8:          **else**
9:              $D_h \leftarrow D_h \cup g(\boldsymbol{\theta}_{h-1}, \boldsymbol{p}_{i_j}^c + \mathcal{N}(0, \sigma^2))$   ▷ Synthesize data samples from centroids with Gaussian noise
10:          **end if**
11:      **end for**
12:      Copy $f(\boldsymbol{\theta}_{h-1})$ and $g(\boldsymbol{\theta}_{h-1})$ to $f(\boldsymbol{\theta}_h), g(\boldsymbol{\theta}_h)$
13:      Calculate unaligned $\{\boldsymbol{p}_{h_j}\} = \operatorname{avg}(f(\boldsymbol{\theta}_{h-1}, T_h))$
14:      Transmit unaligned $\{\boldsymbol{p}_{h_j}\}$ to the server
15:      **for** $\omega = 1$ to $\Omega$ **do**   ▷ For each communication round
16:          **for** $e = 1$ to $E$ **do**   ▷ For each epoch
17:              **for** $b = 1$ to $B$ **do**   ▷ For each minibatch
18:              Minimize the following loss:
19:              $\boldsymbol{L}(D_h, g(\boldsymbol{\theta}_h, f(\boldsymbol{\theta}_h, D_h)), f(\boldsymbol{\theta}_h, D_h)) +$
20:              $\| f(\boldsymbol{\theta}_{h-1}, D_h) - f(\boldsymbol{\theta}_h, D_h) \| +$
21:              $\| g(\boldsymbol{\theta}_{h-1}, f(\boldsymbol{\theta}_{h-1}, D_h)) - g(\boldsymbol{\theta}_h, f(\boldsymbol{\theta}_h, D_h)) \|$
22:              Using an optimizer to obtain $f(\boldsymbol{\theta}_h^*), g(\boldsymbol{\theta}_h^*)$
23:              **end for**
24:          **end for**
25:      **end for**
26:      Update memory: $\mathcal{M}_h \leftarrow f(\boldsymbol{\theta}_h, g(\boldsymbol{\theta}_{h-1}, \mathcal{M}_{h-1}))$   ▷ Re-encode using new encoder and old decoder outputs
27:      Remove $f(\boldsymbol{\theta}_{h-1}), g(\boldsymbol{\theta}_{h-1})$
28:      $\mathcal{M}_h \leftarrow \operatorname{random}(T_h)$ ▷ Randomly sample from the current task's data to populate the memory
29: **end for**

---

Table 2: HR versus exemplar-based and exemplar-free baselines. Accuracies and SEMs for 10 runs.

| Methods/Benchmarks | CIFAR-100 | | ImageNet-Subset | | TinyImageNet |
|---|---|---|---|---|---|
| FCIL Configuration | 10/10/50/5 | 20/5/50/5 | 10/20/100/10 | 20/10/100/10 | 10/5/300/30 |
| Exemplar-based Approaches (20 exemplars per class) | | | | | |
| iCaRL (Rebuffi et al., 2017) | 50.25 ±0.53 | 47.83 ±0.49 | 46.05 ±0.37 | 42.71 ±0.32 | 47.34 ±0.40 |
| BiC (Wu et al., 2019) | 54.38 ±0.49 | 51.03 ±0.43 | 48.83 ±0.41 | 45.02 ±0.36 | 48.79 ±0.43 |
| PODNet (Douillard et al., 2020) | 58.97 ±0.47 | 54.09 ±0.40 | 51.38 ±0.46 | 48.91 ±0.44 | 53.12 ±0.47 |
| DDE (Hu et al., 2021) + iCaRL | 56.46 ±0.50 | 54.12 ±0.48 | 52.01 ±0.40 | 48.31 ±0.40 | 53.06 ±0.41 |
| GeoDL + iCaRL | 60.73 ±0.57 | 56.98 ±0.54 | 49.89 ±0.40 | 47.11 ±0.35 | 50.70 ±0.41 |
| SS-IL (Ahn et al., 2021) | 52.97 ±0.52 | 50.23 ±0.47 | 45.49 ±0.43 | 41.49 ±0.39 | 45.08 ±0.32 |
| GLFC (Dong et al., 2022) | 61.83 ±0.59 | 57.09 ±0.51 | 53.83 ±0.46 | 49.06 ±0.41 | 54.23 ±0.57 |
| Data-Free Approaches | | | | | |
| FedFTG (Zhang et al., 2022) | 46.86 ±0.49 | 43.01 ±0.43 | 42.89 ±0.31 | 39.08 ±0.27 | 43.23 ±0.39 |
| FedSpace (Shenaj et al., 2023) | 45.73 ±0.51 | 41.57 ±0.45 | 40.69 ±0.24 | 38.28 ±0.31 | 42.71 ±0.32 |
| TARGET (Zhang et al., 2023) | 48.38 ±0.42 | 42.79 ±0.39 | 41.04 ±0.25 | 39.92 ±0.34 | 45.13 ±0.42 |
| MFCL (Babakniya et al., 2024) | 50.03 ±0.33 | 44.88 ±0.30 | 43.71 ±0.38 | 40.04 ±0.36 | 46.71 ±0.47 |
| Hybrid Approaches (200 latent exemplars per class) | | | | | |
| REMIND (Hayes et al., 2020) | 62.29 ±0.52 | 58.22 ±0.42 | 53.80 ±0.47 | 49.51 ±0.45 | 54.79 ±0.60 |
| REMIND+ (Wang et al., 2021) | 63.47 ±0.56 | 59.61 ±0.44 | 54.79 ±0.52 | 50.23 ±0.49 | 55.92 ±0.57 |
| i-CTRL (Tong et al., 2022) | 63.31 ±0.49 | 58.93 ±0.40 | 54.55 ±0.38 | 49.92 ±0.36 | 55.23 ±0.54 |
| Our Proposed Hybrid Approach (200 latent exemplars per class) and Ablation Study | | | | | |
| HR | 65.84 ±0.57 | 60.48 ±0.47 | 56.48 ±0.38 | 52.35 ±0.31 | 57.86 ±0.59 |
| HR-mini w 10× less memory | 60.42 ±0.49 | 56.81 ±0.44 | 53.04 ±0.36 | 48.74 ±0.29 | 54.02 ±0.53 |
| HR w/o Latent Exemplars | 52.27 ±0.32 | 45.18 ±0.29 | 47.85 ±0.31 | 43.62 ±0.27 | 48.95 ±0.49 |
| HR w Perfect Exemplars | 66.53 ±0.45 | 62.37 ±0.40 | 57.65 ±0.32 | 54.46 ±0.27 | 59.25 ±0.51 |
| HR w/o KD | 64.07 ±0.52 | 58.13 ±0.41 | 55.71 ±0.34 | 50.38 ±0.29 | 54.80 ±0.55 |
| HR w/o Global Replay | 51.76 ±0.24 | 44.38 ±0.23 | 46.27 ±0.25 | 41.09 ±0.21 | 46.95 ±0.42 |
| HR w RFA | 65.73 ±0.60 | 60.50 ±0.48 | 56.44 ±0.39 | 52.35 ±0.33 | 57.81 ±0.63 |

We benchmark HR against a diverse set of baselines that can be categorized into three approaches: data-based, data-free, and hybrid approaches. Following Dong et al. (2022) we include iCaRL (Rebuffi et al., 2017), BiC (Wu et al., 2019), PODNet (Douillard et al., 2020), DDE (Hu et al., 2021)+iCaRL, GeoDL (Simon et al., 2021) + iCaRL, SS-IL (Ahn et al., 2021), and GLFC (Dong et al., 2022) as the data-based approaches. For data-free approaches, we include FedFTG (Zhang et al., 2022), FedSpace (Shenaj et al., 2023), TARGET (Zhang et al., 2023), and MFCL (Babakniya et al., 2024). And finally, for hybrid approaches, REMIND (Hayes et al., 2020), REMIND+ (Wang et al., 2021), and i-CTRL (Tong et al., 2022) are considered.

**Results** from Table 2 show that exemplar-based approaches generally outperform model-based (data-free) approaches, particularly in scenarios involving longer task sequences. This is because exemplar-based methods can directly use stored exemplars to mitigate forgetting, ensuring more stable and consistent performance across tasks. However, these approaches often come with significant memory costs, which can be impractical for FCIL applications. In contrast, hybrid approaches not only match but often exceed the performance of exemplar-based methods. Their advantage lies in the ability to store compressed or encoded exemplars instead of raw data, enabling up to 10 times more exemplar storage within the same memory budget.

In Table 2, our proposed HR approach, even without latent exemplars (denoted as *HR-mini w* 10× *less memory*), surpasses existing data-free methods by relying solely on synthetic data generation to address forgetting (Line 9 of Algorithm 2). When HR operates in its default mode, using latent exemplars (Line 7 of Algorithm 2), it significantly outperforms exemplar-based baselines, not only in short task sequences but also in longer ones. We attribute this superior performance to HR's strong mathematical foundations leading to a well-justified approach, which ensures effective mitigation of both local and global forgetting.

**Ablation** results are provided in the lowest section of Table 2 where the first scenario labeled *HR-mini w* 10× *less memory* reports the performance of our approach under a significantly reduced memory budget. This scenario demonstrates the potential of HR to perform on par with exemplar-based approaches, despite using 10 times less memory. *HR w/o Latent Exemplars* explores a truly model-based and data-free variant of our approach. This model-based variant of HR outperforms the data-free baselines considerably, proving the robustness of HR even without relying on stored latent exemplars.

*HR w Perfect Exemplars* investigates the potential performance drop caused by the compression and storage of exemplars in the latent space. Interestingly, the results show that the performance difference between the storage of perfect exemplars and latent/encoded exemplars is negligible. This aligns with recent studies in (Van de Ven et al., 2020), suggesting that, for overcoming forgetting (as opposed to learning), the image quality for replay is not a major factor. Thus, even compressed latent exemplars are sufficient for preventing forgetting. *HR w/o KD* examines the impact of removing the knowledge distillation (KD) regularization loss function (Lines 20 and 21 of Algorithm 2). Without KD, we observe a noticeable performance degradation, particularly in long task sequences, primarily due to increased weight drift, which aligns with findings from studies conducted by Rebuffi et al. (2017); Masana et al. (2020). This reinforces the importance of regularization in mitigating forgetting and striking the right balance in the stability-plasticity dilemma.

Another critical aspect of our approach is global replay, which ensures that clients have access to representations of tasks from other clients (Line 9 of Algorithm 2). *HR w/o Global Replay* evaluates the consequences of removing this feature. The results reveal a severe performance drop, which can be attributed to the onset of global forgetting. As expected, without global replay, the system struggles to retain knowledge of tasks learned by other clients, leading to significant degradation in overall performance. Lastly, in the case labeled *HR w RFA*, we replaced the Lennard-Jones potential with the Repulsive Force Algorithm (RFA) (Nazmitdinov et al., 2017) for class embedding alignment. The results indicate that RFA performs comparably to the Lennard-Jones formulation (Jones, 1924).

**Privacy-Performance Trade-off**. Fig. 3 (first row) illustrates the trade-off between privacy and performance across different replay approaches. In exemplar-based methods, performance improves consistently as more memory becomes available to store additional exemplars. By contrast, model-based approaches lack this flexibility, as their performance remains constant regardless of memory size. However, model-based methods offer a significant privacy advantage, as they do not store exact data samples.

Hybrid replay strikes an effective balance between these two approaches: it achieves high performance with low memory consumption while retaining the ability to improve further by storing more exemplars. This can be seen in Fig. 3 (first row) where hybrid replay approaches of HR (ours), REMIND+, and i-CTRL outperform, respectively. In terms of privacy, hybrid replay positions itself between model-based and exemplar-based methods, better than exemplars-based approaches but worse than model-based approaches.

Figure 3 (second row) illustrates the forgetting trends across three FCIL configurations. Hybrid replay approaches, particularly our approach, exhibit the least forgetting. This advantage arises from the ability of hybrid methods to store a greater diversity of exemplars within the same memory budget. The increased memory diversity significantly enhances performance and mitigates forgetting, making hybrid replay a highly effective solution.

Table 3 compares the performance of our hybrid replay with three other exemplar-based replay baselines. It can be seen that for the same given memory and compute budget HR yields greater performance.

## 6 CONCLUSION

In the FCIL literature, two main categories of approaches have been explored to address local and global forgetting: exemplar-based (data-based) and model-based approaches. In this work, we introduced Hybrid Replay (HR), a novel approach that leverages a customized autoencoder for both discrimination and synthetic data generation. By combining latent exemplar replay with synthetic data generation, HR effectively mitigates both local and global forgetting. Furthermore, we presented a mathematical framework to formalize the challenges of local and global forgetting and demonstrated how HR addresses these issues in principle. Extensive experiments across diverse baselines, benchmarks, and ablation studies confirmed the effectiveness of the proposed approach.

*Limitations of HR and potential directions for future work*. Since, our proposed FCIL approach, named HR, heavily relies on data generation based on latent exemplars and class centroids, it requires careful consideration of the decoder's design to strike the right balance between realism and privacy. If the decoder lacks sufficient representational capacity, it may fail to generate realistic images necessary for effective replay. Conversely, if the decoder is overly complex, it could inadvertently

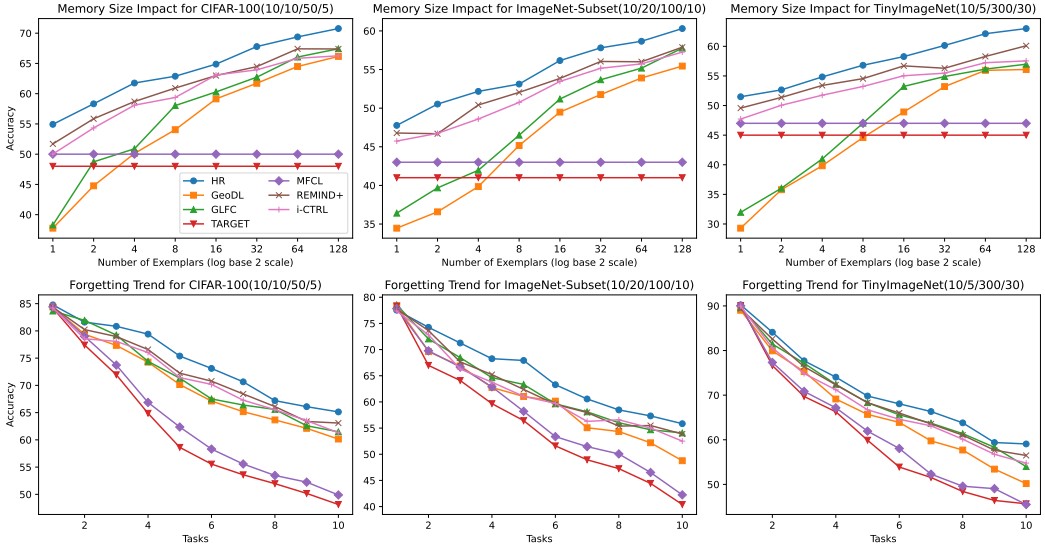

Figure 3: In the first row, the table reports the impact of the memory size on the final accuracy for the three FCIL configurations. In the second row, the forgetting trends are shown for HR and various baselines. Hybrid replay consistently outperforms the model-based or data-based approaches.

Table 3: Performances for a given memory and compute budgets for two FCIL configurations.

| Strategies | Benchmarks | # Exemplars | Memory | | # Epochs | WC Time | Performance |
|---|---|---|---|---|---|---|---|
| | | | Decoder | Exemplar | | | |
| HR | CIFAR-100$(10/10/50/5)$ | 150 (latent) | 1.4M | 4.6M | 50 | 682min | **65.43** $\pm0.93$ |
| | ImageNet-Subset$(10/20/100/10)$ | 190 (latent) | 1.8M | 40.54M | 70 | 974min | **56.09** $\pm0.64$ |
| GeoDL | CIFAR-100$(10/10/50/5)$ | 20 (raw) | - | 6M | 60 | 763min | 60.12 $\pm0.91$ |
| | ImageNet-Subset$(10/20/100/10)$ | 20 (raw) | - | 42.34M | 80 | 1146min | 49.23 $\pm0.62$ |
| SS-IL | CIFAR-100$(10/10/50/5)$ | 20 (raw) | - | 6M | 60 | 725min | 52.81 $\pm0.74$ |
| | ImageNet-Subset$(10/20/100/10)$ | 20 (raw) | - | 42.34M | 80 | 1096min | 45.67 $\pm0.63$ |
| GLFC | CIFAR-100$(10/10/50/5)$ | 20 (raw) | - | 6M | 60 | 788min | 61.70 $\pm0.79$ |
| | ImageNet-Subset$(10/20/100/10)$ | 20 (raw) | - | 42.34M | 80 | 1259min | 53.83 $\pm0.55$ |

memorize training samples, raising potential privacy concerns. This trade-off is particularly important because, for replay purposes, unlike for learning, generating high-quality samples is not necessary and a larger decoder would inflict a higher computational load during training (but not inference because the classification is only done via the encoder). Future work could explore alternative decoder architectures that optimize this balance, tailoring the trade-off between computational efficiency, replay performance, and privacy to the requirements of the target application.

## ACKNOWLEDGMENT

This research was partly funded by the Natural Sciences and Engineering Research Council of Canada (NSERC) and the Rogers Cybersecure Catalyst Fellowship Program.

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
