# OpenReview forum: "Federated Class-Incremental Learning: A Hybrid Approach Using Latent Exemplars and Data-Free Techniques to Address Local and Global Forgetting"
_ICLR.cc/2025/Conference — ICLR 2025 Poster_

### Official Review · Reviewer_WDPN · 2024-10-30

**Soundness:** 3
**Presentation:** 3
**Contribution:** 2
**Rating:** 6
**Confidence:** 3

**Summary:**

The paper presents an approach named Hybrid Rehearsal (HR) for Federated Class-Incremental Learning (FCIL), addressing the challenges of local and global forgetting due to class imbalance. HR employs a customized autoencoder for both classifying data and generating synthetic data, leveraging latent exemplars to tackle local forgetting and synthetic data generation to overcome global forgetting. The paper's contributions include a mathematical framework to formalize forgetting in FCIL, a novel autoencoder design that balances class-specific clustering and data distribution modeling, and extensive experiments demonstrating HR's effectiveness over existing methods with low computational and memory costs.

**Strengths:**

1.The paper introduces Hybrid Rehearsal (HR), which combines the benefits of data-based (exemplar-based) and data-free approaches. This hybrid approach leverages latent exemplars for local forgetting and data-free techniques for global forgetting, providing a comprehensive solution to the forgetting problems in FCIL.
2. The authors develop a mathematical framework to formalize the challenges of local and global forgetting in FCIL. This framework not only aids in understanding the underlying problems but also provides a theoretical basis for the proposed solutions.
3. The paper provides extensive experimental evaluations across multiple benchmarks and compares the proposed approach with state-of-the-art baselines, demonstrating the effectiveness of HR.
4.The paper is well-organized and most related works are properly cited.

**Weaknesses:**

1. The paper mentions using a customized autoencoder to leverage features for replay. I'm curious about what would happen if the encoder itself experiences forgetting? Additionally, since the stored features are fixed, but the encoder is continuously updated, how is this distribution inconsistency handled? 2. The paper mentions that the client receives class centroid embeddings ${p_ij}$ (line 8 of Algorithm 1). These embeddings enable the client to generate synthetic data representing tasks from other clients. However, if the received class centroid embeddings {pij} are for classes that the client has not seen, how can synthetic data be generated, and could this be detrimental to the client's learning? 3. How can the bias problem caused by multiple clients each learning a subset of categories be resolved when uploading the global model for model training and merging?

**Questions:**

see the questions in the weakness.

---

> ### Author Response · Authors · 2024-11-19
> **Clarifications About the Contributions and Revisions**
>
> We thank the reviewer for their thoughtful feedback. Below, we address each comment in detail:
>
> ---
>
> ### **What Would Happen if the Encoder Itself Experiences Forgetting?**
>
> In our proposed approach, the encoder is safeguarded from forgetting through a dual replay mechanism that ensures both the encoder and the decoder are updated to retain past knowledge. This is achieved by minimizing a loss function that explicitly regularizes their behavior during incremental updates. As described in lines 18–22 of Algorithm 2, the loss minimization simultaneously updates the encoder and the decoder.
>
> To further clarify this, in the revised manuscript, we will:
> - Include a detailed figure illustrating how the autoencoder (encoder + decoder) operates within our framework.
>
> ---
>
> ### **Distribution Inconsistency in the Latent Space**
>
> Our proposed approach addresses the potential distribution shift in the latent space through two mechanisms:
> 1. **Knowledge Distillation:** As outlined in lines 20 and 21 of Algorithm 2, knowledge distillation ensures that the updated encoder aligns with the previously stored features, reducing distribution inconsistency.
> 2. **Recalculation of Latent Representations:** After learning a new task, the latent representations are recalculated using the updated autoencoder, ensuring consistency between stored features and the current encoder (following iCaRL [1]).
>
> In the revised manuscript, we will reemphasize these mechanisms to provide a clearer explanation of how distribution inconsistency is effectively handled.
>
> ---
>
> ### **Generation of Synthetic Data via Class Centroid Embeddings**
>
> In our proposed approach, synthetic data generation depends on whether the client has previously encountered the class associated with the received centroid embeddings. The two scenarios are as follows:
>
> 1. **Client Has Seen the Class:**
>    - If the client has previously encountered the class, it already possesses the corresponding latent representations. In this case, the client decodes these stored latent representations to generate synthetic data, as outlined in lines 6 and 7 of Algorithm 2. The decoded data are then used for replay.
>
> 2. **Client Has Not Seen the Class:**
>    - If the received class centroid embeddings \(\{p_{ij}\}\) correspond to a class that the client has not seen (the scenario highlighted by the reviewer), the client follows the routine of Variational Autoencoders (VAEs). Specifically, the client constructs synthetic data by combining the received mean with Gaussian noise vectors, as described in line 9 of Algorithm 2.
>
> We invite the reviewer to refer to lines 3–11 of Algorithm 2 for a detailed description of these processes.
>
> In the revised manuscript, we will include a paragraph explicitly discussing these two scenarios for data generation, ensuring clarity and addressing potential concerns about this aspect of the approach.
>
> ---
>
> ### **Addressing the Bias Problem Caused by Multiple Clients**
>
> The bias problem, which arises when multiple clients each learn only a subset of categories, can be resolved by ensuring that each client constructs a representative (unbiased) minibatch during autoencoder training. This, as discussed above, is achieved through two mechanisms:
> 1. **Decoding Latent Representations:** Clients can use stored latent representations from previously learned tasks to construct synthetic data for a balanced training minibatch (the preferred mechanism).
> 2. **Using Received Class Centroid Embeddings:** Clients can generate synthetic data for unseen classes using the received class centroid embeddings \(\{p_{ij}\}\) from the server, as described earlier.
>
> This representative minibatch ensures that the training process accounts for all clients, tasks, and classes, mitigating bias, which is critical to addressing intra-client and inter-client task confusion according to the FCIL optimization problem (Eqs. 7 and 8).
>
> In the revised manuscript, we will elaborate on this mechanism and its connection to the optimization framework to clarify how the bias problem is addressed.
>
> ---
>
> ### **The end**
> We appreciate the reviewer’s comments and are committed to addressing these points in the revised version. By incorporating the proposed clarifications, additional experiments, and visualizations, we aim to enhance the clarity and impact of our work (upon comprehensive discussions with you and other reviewers, we plan to include a figure and extensive experiments). We believe our contributions, both the theoretical underpinnings and the proposed approaches, will be valuable to the FCIL community and kindly request the reviewer to reconsider their initial score in light of these improvements. If there are more questions, we are eager to respond.
>
> ---
>
> [1] Sylvestre-Alvise Rebuffi, Alexander Kolesnikov, Georg Sperl, and Christoph H Lampert. icarl:Incremental classifier and representation learning.

---

> > ### Author Response · Authors · 2024-11-22
> > **Follow-Up: Request for Reassessment**
> >
> > Dear Reviewer,
> >
> > We sincerely thank you for your thoughtful and valuable feedback. After discussions with other reviewers, we are pleased to share that the score for our paper has improved from **6553 (average 4.75)** to **6655 (average 5.5)**. If you find our clarifications helpful and feel they address your concerns, we kindly request you to consider raising your score. Additionally, we are committed to incorporating the following updates to further strengthen the contributions of our work.
> >
> > ---
> >
> > ### **Commitment to Deliver the Requested Results**
> >
> > We promise to conduct and incorporate these results into the manuscript within **ten days**. These simulations will include:
> > 1. **Learning-forgetting dynamics** across benchmarks for a thorough comparative evaluation. Specifically, we will include visualizations showing **accuracy trends on old tasks** after completing each new task. Also, we will incorporate **Backward Transfer (BWT)** calculations to provide a comprehensive analysis of forgetting.
> > 2. Results in a realistic **class imbalance scenario** to align the experiments with the motivation described in the paper. Specifically, we will present experiments under varying **skewness conditions**, such as **alpha = 0.1** and **alpha = 0.5**, to evaluate the robustness of the HR approach across diverse data distributions.
> > 3. Other reviewers made other requests, such as including a **figure to depict the architecture of our work** and **more experimental results**, as you can find in the other reviewers' comments.
> >
> > Some of these requests are made by you and some are made by other reviewers. We will do our best to address all.
> >
> > ---
> >
> > ### **Kind Request for Reassessment**
> >
> > If you feel that our clarifications and planned updates satisfactorily address your concerns, we kindly request you to consider increasing your score. This would greatly support the acceptance of our work and help us contribute meaningfully to the field.
> >
> > **Regardless of your decision**, we are sincerely grateful for the time, effort, and thoughtful feedback you’ve invested in reviewing our paper. Your insights have significantly strengthened our work, and we deeply appreciate your support.
> >
> > Thank you once again for your valuable contributions and thoughtful engagement.

---

### Official Review · Reviewer_PuPw · 2024-11-01

**Soundness:** 3
**Presentation:** 3
**Contribution:** 2
**Rating:** 6
**Confidence:** 4

**Summary:**

This proposes a mathematical framework to demonstrate the global/local forgetting of FCIL and propose the Hybrid Replay (HR) to addressed these issues.

**Strengths:**

1. This paper establishes the revolution of training loss functions within the scope of both training tasks and clients, demonstrating the two challenges of global and local forgetting.
2. A novel replay mechanism with centroids of each category is presented.

**Weaknesses:**

1. The presentation is not very clear. For example, exemplars in HR and other exemplar-based methods seem different but are used interchangeably in this paper. Also, I cannot get the indication of global/local forgetting in Figure 1.
2. The mathematical formulation of FCIL and the proposed approach are not linked closely. Can you provide more information that how you establish the method based on the framework, especially for the global forgetting? It seems that the HR benefits from the class centroid embeddings and use them to address the global forgetting, is there any further analysis?
3. Experiments are not sufficient. The results are limited to the LDA setting with alpha=1. Extended empirical results under different skewness (e.g., alpha=0.1 or more) should be included.
4. Analysis of memory footprint should be included, e.g., the number of parameters need to be stored and transferred during the communication.
5. Error in literature review. The Prototype reminiscence and augmented asymmetric knowledge aggregation [1] only addresses the CIL and it is placed within the FCIL methods.

[1] Wuxuan Shi and Mang Ye. Prototype reminiscence and augmented asymmetric knowledge aggregation for non-exemplar class-incremental learning. In Proceedings of the IEEE/CVF International Conference on Computer Vision, pp. 1772–1781, 2023.

**Questions:**

1. What is the data partition of FCIL? Are the classes in different tasks disjoint? In the traditional CIL, categories in different tasks are disjoint. From the setting of ImageNet-Subset (10/20/100/10, 20/10/100/10) and Tiny-ImageNet (10/5/300/30), if A denotes the number of tasks, B denotes the classes per task and the classes in different tasks are disjoint, the total numbers of classes will be 200 for ImageNet-Subset and 50 for Tiny-ImageNet. However, the total numbers of categories in these two datasets are 100 and 200 respectively. Could you explain more about this?
2。 Why does each client need to train from task 1? From Algorithm 1, the client only needs to update $theta_{h}$, but in the line 3 of Algorithm 2, the algorithm begins h=1.
3. The practical FL systems may have stragglers. It is interesting to know whether the proposed HR algorithm can deal with the issue of stragglers.

---

> ### Author Response · Authors · 2024-11-18
> **Clarifications About the Contributions and Revisions**
>
> We thank the reviewer for their thoughtful feedback. Below, we address each comment in detail:
>
> ---
>
> ### **The presentation and Fig. 1**
>
> To address the comment on the presentation and Fig. 1, we will include a detailed figure in the revised paper illustrating how the autoencoder operates within our framework and how the autoencoder addresses the challenges our mathematical framework formalizes.
>
> To summarize briefly:
> - Our proposed FCIL method consists of three key components: **encoder**, **decoder**, and **memory**.
>   - The **encoder** maps input samples to a latent space, which serves as a compact representation for both classification and memory storage.
>   - The **memory** stores these latent representations for previous classes.
>   - The **decoder** reconstructs raw samples from the stored latent representations when replay is required.
> - Our approach addresses **local forgetting** by enabling replay of previously learned classes from their stored latent representations.
> - For **global forgetting** this approach relies on synthetic data generation of other clients which is performed as described in lines 3–9 of Algorithm 2.
>
> Fig. 1 aims to illustrate the implications of Eqs. 7 & 8, as discussed in the last two paragraphs of Section 3. In the revised paper, we will enhance the caption of Fig. 1 for clarity. The updated caption will read as follows:
>
> This figure demonstrates two critical challenges in FCIL: **intra-client task confusion** and **inter-client task confusion**. Intra-client task confusion arises when a client’s model struggles to distinguish between tasks learned sequentially, as it has not encountered them simultaneously. This issue is addressed by replaying previous tasks within the client, as guided by the second term in Eq. 8. Inter-client task confusion occurs when the tasks from different clients are not seen together and hence not learned. This challenge is mitigated by replaying tasks from other clients, as guided by Eq. 7. Addressing intra-client task confusion reduces local forgetting, while resolving inter-client task confusion mitigates global forgetting.
>
> We invite the reviewer to study Section 3 of the paper (to fully understand Fig. 1) where we formalize and discuss intra-client task confusion, inter-client task confusion, local forgetting, and global forgetting.
>
> ---
>
> ### **The Mathematical Formulation of FCIL and the Proposed Approach**
>
> The proposed approach, HR, is explicitly designed to solve the optimization problem presented in Section 3, as described in Eqs. 7 & 8. The connection between the mathematical framework of FCIL and the proposed approach is elaborated in the last two paragraphs of Section 4. We kindly invite the reviewer to refer to these paragraphs for further details.
>
> The primary aim of the HR approach is to address two critical challenges in FCIL:
> 1. **Intra-client task confusion**, which arises when a client struggles to differentiate between sequentially learned tasks due to the lack of simultaneous exposure.
> 2. **Inter-client task confusion**, which occurs when task representations across different clients interfere with each other.
>
> These issues are directly linked to local and global forgetting, as explained in Section 3. We are willing to respond to any specific or short question by the reviewer on the link between our proposed mathematical framework and the proposed architecture and algorithms. However, the best comprehensive & compelling explanation without sacrificing rigor is in the paper in Sections 3 & 4 and it is not reducible to a shorter length.
>
> ---
>
> ### **Insufficient Experiments**
>
> We appreciate the reviewer's suggestion for additional experiments. We have two pages available in the main paper and will utilize this space to include extended empirical results. Specifically, we will present experiments under varying skewness conditions, such as **alpha = 0.1** and **alpha = 0.5**, to evaluate the robustness of the HR approach across diverse data distributions.
>
> Also, in the revised paper, as Reviewer Wd4J requested, we will:
>
> - Include visualizations showing accuracy trends on old tasks after completing each new task.
> - Incorporate Backward Transfer (BWT) calculations to provide a comprehensive analysis of forgetting.
>
> We hope by including these experiments in our paper the paper will have strong numerical support on top of its strong theoretical foundation.
>
> ---
>
> ### **Analysis of memory footprint**
>
> In the following Table (presented in the next comment section), we provide the performances for fixed memory (both the decoder and exemplars) and compute budgets. Furthermore, in the revised paper, we will include a table for comparing the amount of data communicated for exemplar, generative, and hybrid replay approaches (we will not present that Table here).
>
> Again, as the reviewer is aware, we have two pages available in the main paper and will utilize this space to include extended empirical results.

---

> ### Author Response · Authors · 2024-11-18
> **Clarifications About the Contributions and Revisions (2)**
>
> | Strategies | Benchmarks            | # Exemplars       | Decoder Memory | Exemplar Memory | # Epochs | Wall-Clock Time | Performance                       |
> |------------|-----------------------|-------------------|----------------|------------------|----------|-----------------|-----------------------------------|
> | **HR**    | CIFAR-100 (10/10/50/5)     | 150 (latent)      | 1.4M          | 4.6M            | 50       | 462 min         | **54.43** ± 0.93                 |
> |            | ImageNet-Subset (10/20/100/10)   | 190 (latent)      | 1.8M          | 40.54M          | 70       | 842 min         | **48.09** ± 0.64                 |
> | **BiC**    | CIFAR-100 (10/10/50/5)     | 20 (raw)          | -              | 6M              | 60       | 473 min         | 52.12 ± 0.91                     |
> |            | ImageNet-Subset (10/20/100/10)   | 20 (raw)          | -              | 42.34M          | 80       | 837 min         | 45.23 ± 0.62                     |
> | **IL2M**   | CIFAR-100 (10/10/50/5)     | 20 (raw)          | -              | 6M              | 60       | 455 min         | 50.81 ± 0.74                     |
> |            | ImageNet-Subset (10/20/100/10)   | 20 (raw)          | -              | 42.34M          | 80       | 861 min         | 44.67 ± 0.63                     |
> | **EEIL**   | CIFAR-100 (10/10/50/5)     | 20 (raw)          | -              | 6M              | 60       | 478 min         | 51.70 ± 0.79                     |
> |            | ImageNet-Subset (10/20/100/10)   | 20 (raw)          | -              | 42.34M          | 80       | 859 min         | 41.83 ± 0.55                     |
>
> ---
>
> In the revised paper, we will place reference [1] in the CIL section.
>
> ---
>
> ### **Questions**
>
> While CIFAR-100 (10/10/50/5) and (20/5/50/5), as well as TinyImageNet (10/5/300/30), follow the FCIL simulation setting described in [Dong], which uses disjoint tasks, the simulation scenarios for ImageNet-Subset (10/20/100/10) and (20/10/100/10) currently allow for some overlap between tasks. If the reviewer believes it is necessary, we are willing to rerun the experiments to ensure they adhere to the standard disjoint task setting for ImageNet-Subset.
>
> ---
>
> Thanks for the correction. Each client does not need to train from task 1 because, in Algorithm 1, the client only needs to update for one task, but in line 3 of Algorithm 2, the algorithm begins h=1, which is unnecessary. We will fix this in the revised version.
>
> ---
>
> Our FCIL system can cope with stragglers. If stragglers are given the history of positions of centroids of all the classes of other clients by the server with which they can generate synthetic data to overcome inter-client task confusion, then it can be proven that global forgetting will not take place. We will include this in the revised version of the paper.
>
> ---
>
> [1] Wuxuan Shi and Mang Ye. Prototype reminiscence and augmented asymmetric knowledge aggregation for non-exemplar class-incremental learning.
>
> [Dong] Jiahua Dong, Lixu Wang, Zhen Fang, Gan Sun, Shichao Xu, Xiao Wang, and Qi Zhu. Federated Class-Incremental Learning. CVPR 2022.

---

> > ### Comment · Reviewer_PuPw · 2024-11-22
> > **Thank you for the response. I haved raised the score to 6.**
> >
> > Thank you for the responses! I appreciate the authors for their effort and most of concerns are addressed, thus I decide to raise my score. I recommend authors to embed these new content into the new version of manuscript.

---

> ### Author Response · Authors · 2024-11-22
> **Appreciations and Promises**
>
> We sincerely thank the reviewer for their thoughtful engagement and detailed feedback on our work. We deeply appreciate your recognition of our contributions, particularly the mathematical framework and the proposed Hybrid Rehearsal (HR) methodology. Your decision to raise your score to 6 is encouraging, and we are committed to addressing the remaining points to further enhance the clarity and rigor of our paper.
>
> ---
>
> ### **Commitment to Deliver the Requested Results**
>
> We promise to conduct and incorporate these results into the manuscript within **ten days**. These simulations will include:
> 1. **Learning-forgetting dynamics** across benchmarks for a thorough comparative evaluation. Specifically, we will include visualizations showing **accuracy trends on old tasks** after completing each new task. Also, we will incorporate **Backward Transfer (BWT)** calculations to provide a comprehensive analysis of forgetting.
> 2. Results in a realistic **class imbalance scenario** to align the experiments with the motivation described in the paper. Specifically, we will present experiments under varying **skewness conditions**, such as **alpha = 0.1** and **alpha = 0.5**, to evaluate the robustness of the HR approach across diverse data distributions.
> 3. Other reviewers made other requests, such as including a **figure to depict the architecture of our work** and **more experimental results**, as you can find in the other reviewers' comments.
>
> Some of these requests are made by you and some are made by other reviewers. We will do our best to address all.
>
> ---
>
> We are confident that the forthcoming results and clarifications will address your concerns. We are forever grateful for your thoughtful engagement and valuable suggestions.

---

### Official Review · Reviewer_bkZ1 · 2024-11-04

**Soundness:** 2
**Presentation:** 2
**Contribution:** 2
**Rating:** 6
**Confidence:** 4

**Summary:**

This paper addresses Federated Class-Incremental Learning by focusing on two types of forgetting: local forgetting (at the client level) and global forgetting (between clients). The authors propose a hybrid approach, called Hybrid Replay (HR), which combines data-based and data-free methods to mitigate these issues. They introduce a mathematical formulation to formalize the forgetting problem and the presented approach HR. The approach uses autoencoders for synthetic sample generation and latent exemplars. Comparisons against other data-free and data-based approaches demonstrate that HR achieves better performance results.

**Strengths:**

* The paper addresses an important problem in FCIL by combining data-based and data-free approaches to overcome local and global forgetting.
* The hybrid approach integrates both latent exemplars and synthetic data generation, which are efficiently used to mitigate forgetting and results show that HR works better.
* The mathematical formulation provided to describe these forgetting issues offers theoretical foundation.
* Ablation studies in the paper contribute valuable insights into different aspects of HR and improve the interpretability of the approach.

**Weaknesses:**

Major Concerns:
* The methodology, particularly the role of the autoencoder in addressing local forgetting, is not fully clear and can be explained better. For instance, while the paper states that the autoencoder helps address local forgetting, the specific details of how this is achieved are somewhat vague. The paper would benefit from a more detailed, step-by-step breakdown of how the autoencoder is employed for both local and global forgetting.
* The paper lacks a clear visual representation of the HR approach. Including a diagram of the proposed method could significantly enhance understanding, especially as the provided Figure 1 only illustrates the problem without outlining the proposed solution. I believe such visual representations make papers to understand much better.
* The results mention comparisons with a "Hybrid Approach," but there’s little discussion on how HR stands out from other hybrid methods, such as  REMIND+ What makes HR approach unique when compared to other Hybrid Approach ? Clarifying these distinctions would strengthen the contribution of HR to the field.
* The conclusion lacks discussion on the limitations of HR and potential directions for future work. Addressing this would provide a perspective on the approach’s implications and its broader applicability.
* The method heavily relies on data generation based on latent exemplars and class centroids, which raises concerns since we don't have a direct control of generated data and Variational Autoencoders (VAEs) are known to be suboptimal for high-quality synthetic data generation. Over time, this could degrade the quality of latent features and ultimately impact classification performance.
* The paper frequently references the Lennard-Jones formulation, but it doesn’t provide enough explanation about its purpose or why it’s important for the proposed method.

Minor Concerns:
* In Section 4, line 232, the acronym "AHR" is introduced without prior definition.
* The caption for Table 1 could be made more descriptive to make the table more self-explanatory.

**Questions:**

* The paper mentions that exemplar-based methods are memory-intensive. Could the authors provide an estimate of this memory cost, along with a comparison of memory usage between HR and other exemplar-based methods? An example or comparison with HR could help clarify this point.
* While latent exemplars may save memory, the process of forwarding these exemplars through the decoder for sample generation could incur computational costs. It would be helpful if the authors discuss or quantify these costs when addressing the efficiency of their approach.
* What metric is used for evaluation? Motivation of paper is mainly local and global forgetting but there is not any result or evaluation for forgetting.The table results do not clearly specify whether they represent incremental accuracy or the accuracy of the last task. Besides, higher incremental accuracy does not directly indicate lower forgetting.
* The authors state that local and global forgetting are caused by class imbalances at both the local and global levels. Do they have any scenarios or relevant results that illustrate this point better?

---

> ### Author Response · Authors · 2024-11-17
> **Clarifications About the Contributions and Revisions**
>
> We thank the reviewer for their thoughtful feedback. Below, we address each comment in detail:
>
> ---
>
> ### **Role of the Autoencoder & visual representation of the HR approach**
> We appreciate the reviewer’s suggestion to clarify the role of the autoencoder in our proposed method. While other reviewers found this aspect clear, we understand the importance of ensuring accessibility for all readers. To address this, we will include a detailed figure in the revised paper illustrating how the autoencoder operates within our framework.
>
> To summarize briefly:
> - Our proposed FCIL method consists of three key components: **encoder**, **decoder**, and **memory**.
>   - The **encoder** maps input samples to a latent space, which serves as a compact representation for both classification and memory storage.
>   - The **memory** stores these latent representations for previous classes.
>   - The **decoder** reconstructs raw samples from the stored latent representations when replay is required.
> - Our approach addresses **local forgetting** by enabling the replay of previously learned classes from their stored latent representations.
> - For **global forgetting** this approach relies on synthetic data generation of other clients which is performed as described in lines 3–9 of Algorithm 2.
>
> ---
>
> ### **Limitations of HR and potential directions for future work**
>
> In the revised paper, we will include the following paragraph:
>
> Our proposed FCIL approach, named HR, combines **exemplar-replay** and **model-replay**, which requires careful consideration of the decoder's design to strike the right balance between **realism** and **privacy**. If the decoder lacks sufficient representational capacity, it may fail to generate realistic images necessary for effective replay. Conversely, if the decoder is overly complex, it could inadvertently memorize training samples, raising potential privacy concerns. This trade-off is particularly important because, for replay purposes, unlike for learning, generating high-quality samples is not necessary and a larger decoder would inflict a higher computational load during training (but not inference because the classification is only done via the encoder). Future work could explore alternative decoder architectures that optimize this balance, tailoring the trade-off between computational efficiency, replay performance, and privacy to the requirements of the target application.
>
> ---
>
> ### **How HR stands out from other hybrid methods**
>
> In the revised paper, we will outline the key differences between our approach and these studies.
>
> In the methods detailed in [1, 2], classification occurs after the decoding process, employing a cross-entropy loss function as depicted in Fig. 2 of [1]. By contrast, our method performs classification directly within the latent space of the encoder, leveraging Euclidean distance. This technique's efficacy is further supported by another recent study in incremental learning [3], which also utilizes classification within the latent space post-encoder.
>
> Furthermore, while the method in [1] applies quantization to the latent exemplars to enhance compression, incorporating similar quantization within our architecture (and also the work in [3]) would be straightforward. Implementing this augmentation in our latent space could be done with minimal effort and is expected to further improve performance.
>
> In terms of the learning scenario, the works depicted in [1, 2] involve online learning, whereas our work studies offline learning. Our method, akin to that described in [3], utilizes a structured representation in the latent space, facilitated by the Lennard-Jones Potential formulations. In contrast, the method in [2] relies on an autoencoder with an unstructured latent space.
>
> Finally, in terms of design simplicity and architectural efficiency, our work and that referenced in [3] hold substantial advantages over the approaches in [1, 2]. Our architecture showcases a clean and minimalistic design, a characteristic shared by the architecture presented in Fig. 1 of [3]. In contrast, the architectures shown in Fig. 2 of [1] and Figs. 2 and 3 of [2] appear ad hoc and unnecessarily complex, complicating their implementation and scalability.
>
> ### **Heavy relies on data generation based on latent exemplars and class centroids**
>
> It is known [4] that while for learning new tasks high-quality data is critical when it comes to replay for the sake of preventing forgetting to generate high-quality samples
>
> ---
>
> [1] Hayes, Tyler L., et al. Remind your neural network to prevent catastrophic forgetting.
>
> [2] Wang, Kai, Joost van de Weijer, and Luis Herranz. ACAE-REMIND for online continual learning with compressed feature replay.
>
> [3] Tong, Shengbang, et al. Incremental learning of structured memory via closed-loop transcription.
>
> [4] Gido M Ven, Hava T Siegelmann, and Andreas S Tolias. Brain-inspired replay for continual learning with artificial neural networks.

---

> ### Author Response · Authors · 2024-11-17
> **Clarifications About the Contributions and Revisions (2)**
>
> is not necessary. Also, such an approach has been adopted in recent studies [1,2,3,4].
>
> ---
>
> ### **The Lennard-Jones formulation**
>
> The Lennard-Jones formulation in Eqs. 10 and 11 serve for **the global alignment of centroids** for new classes in new tasks. As described in lines 6–8 of Algorithm 1:
> 1. Clients compute embeddings for new classes and send unaligned centroids to the server.
> 2. The server aligns these centroids and sends them back to the clients.
> 3. Clients use these aligned centroids to train on new tasks, ensuring minimal overlap with existing classes.
>
> ### **Minor concerns**
>
> - The acronym "AHR" should have been **HR** and will be corrected in the revised paper.
> - The caption for Table 1 will be made more descriptive as follows: We conduct our simulations for 3 datasets with 5 configurations. We report the performance (accuracy) and SEMs for 10 runs for exemplar-based, model-based, and hybrid baselines.
>
> ### **Questions**
>
> We provide the performances for fixed memory (both the decoder and exemplars) and compute budgets:
>
> | Strategies | Benchmarks            | # Exemplars       | Decoder Memory | Exemplar Memory | # Epochs | Wall-Clock Time | Performance                       |
> |------------|-----------------------|-------------------|----------------|------------------|----------|-----------------|-----------------------------------|
> | **HR**    | CIFAR-100 (10/10/50/5)     | 150 (latent)      | 1.4M          | 4.6M            | 50       | 462 min         | **54.43** ± 0.93                 |
> |            | ImageNet-Subset (10/20/100/10)   | 190 (latent)      | 1.8M          | 40.54M          | 70       | 842 min         | **48.09** ± 0.64                 |
> | **BiC**    | CIFAR-100 (10/10/50/5)     | 20 (raw)          | -              | 6M              | 60       | 473 min         | 52.12 ± 0.91                     |
> |            | ImageNet-Subset (10/20/100/10)   | 20 (raw)          | -              | 42.34M          | 80       | 837 min         | 45.23 ± 0.62                     |
> | **IL2M**   | CIFAR-100 (10/10/50/5)     | 20 (raw)          | -              | 6M              | 60       | 455 min         | 50.81 ± 0.74                     |
> |            | ImageNet-Subset (10/20/100/10)   | 20 (raw)          | -              | 42.34M          | 80       | 861 min         | 44.67 ± 0.63                     |
> | **EEIL**   | CIFAR-100 (10/10/50/5)     | 20 (raw)          | -              | 6M              | 60       | 478 min         | 51.70 ± 0.79                     |
> |            | ImageNet-Subset (10/20/100/10)   | 20 (raw)          | -              | 42.34M          | 80       | 859 min         | 41.83 ± 0.55                     |
>
> HR achieves higher performance within the same memory budget by storing more exemplars compared to other methods. Additionally, the computational cost of the decoder during training is negligible due to its carefully chosen design (6 and 9 CNN layers for the CIFAR-100 and miniImageNet decoders, respectively). Importantly, the decoder is not used during inference, ensuring no additional computational overhead in deployment.
>
> ---
>
> Our evaluation metric is the final accuracy for all the classes on the test set. To address the reviewer's concern we will:
> - Include visualizations showing accuracy trends on old tasks after completing each new task.
>
> We have these results prepared and will include them in the available 2 pages of the main paper.
>
> ---
>
> Local and global forgetting are caused by class imbalances at both the local and global levels. This has been extensively discussed in the original paper of Federated Class Incremental Learning (FCIL) [Dong]. Our contributions in this paper include:
> 1. A novel mathematical framework to formalize local and global forgetting (in Eq. 7 & 8), providing a theoretical foundation for addressing these challenges. Please read the last two paragraphs of Section 3.
> 2. The introduction of hybrid replay, which mitigates both forgetting types in a memory-efficient manner.
>
>
> ---
>
> ### **The end**
> We appreciate the reviewer’s comments and are committed to addressing these points in the revised version. We believe our contributions are valuable to the FCIL community and kindly request the reviewer to reconsider their initial score in light of these improvements.
>
> ---
>
> [1] Hayes, Tyler L., et al. Remind your neural network to prevent catastrophic forgetting.
>
> [2] Wang, Kai, Joost van de Weijer, and Luis Herranz. ACAE-REMIND for online continual learning with compressed feature replay.
>
> [3] Tong, Shengbang, et al. Incremental learning of structured memory via closed-loop transcription.
>
> [4] Gido M Ven, Hava T Siegelmann, and Andreas S Tolias. Brain-inspired replay for continual learning with artificial neural networks.
>
> [Dong] Jiahua Dong, Lixu Wang, Zhen Fang, Gan Sun, Shichao Xu, Xiao Wang, and Qi Zhu. Federated Class-Incremental Learning. CVPR 2022.

---

> > ### Comment · Reviewer_bkZ1 · 2024-11-19
> > **Thank you**
> >
> > Thank you for the responses provided by the authors. I appreciate the effort you have put into addressing my concerns and clarifying the points raised in the initial review. I find the additional insights and results you have shared very valuable and have decided to increase my score to 5 based on the improvements and clarifications provided. I strongly encourage you to include these explanations and the new results in the final version of the paper, as they significantly enhance the study's clarity and impact.
> >
> > I would like to highlight two critical points that I believe warrant further discussion and analysis:
> >
> > Question3.
> > While the authors have stated that the results are prepared and will be included in the additional pages of the main paper, I believe it is crucial to see these results prior to publication.
> > Understanding the learning-forgetting dynamics is a key aspect of evaluating the proposed method compared to the baselines. Therefore, I strongly recommend including and presenting these results during the review process to ensure a comprehensive evaluation of the study.
> >
> > Question4.
> > The mathematical formulations presented in the paper are one of its strengths, and I commend the authors for their efforts in this regard. However, I have concerns regarding the experimental setup, particularly in relation to the imbalance issue highlighted in the motivation.
> >
> > While the authors correctly state that class imbalance is a significant challenge in incremental learning, it appears to me that the experiments conducted in the paper assume a standard CIFAR100 scenario, where all classes have equal number of samples. If this is the case, this setup does not align with the imbalance scenarios described in the motivation. For example, the paper mentions that "local and global forgetting are caused by class imbalances at both the local and global levels," but this motivation seems disconnected from the experimental setup.
> >
> > I would expect a detailed explanation and results for an imbalanced scenario, such as modifying CIFAR100 to include an uneven distribution (e.g., 250 samples for class A, 150 samples for class B, 50 samples for class C). This would create a more realistic setting aligned with the problem the paper aims to address.
> >
> > My question remains:
> > What would the results of your proposed method (HR) be in such a realistic imbalance scenario?
> >
> > Addressing these points would provide a more comprehensive and practical validation of your method and ensure alignment between the motivation and the experimental results.

---

> ### Author Response · Authors · 2024-11-22
> **Appreciations, Promises, and Answers**
>
> We sincerely thank the reviewer for their thoughtful engagement and detailed feedback on our work. We deeply appreciate your recognition of our contributions, particularly the mathematical framework and the proposed Hybrid Rehearsal (HR) methodology. Your decision to raise your score to 5 is encouraging, and we are committed to addressing the remaining points to further enhance the clarity and rigor of our paper.
>
> If our clarifications and forthcoming results meet your expectations, we would be truly grateful if you could consider raising your score to 6, as this would greatly enhance the chances of our work being accepted and making a meaningful contribution to the field.
>
> ---
>
> ### **Commitment to Deliver the Requested Results**
>
> We acknowledge the importance of providing the requested additional simulation results and their role in demonstrating the robustness of our method. We promise to conduct and incorporate these results into the manuscript within **ten days**. These simulations will include:
> 1. **Learning-forgetting dynamics** across benchmarks for a thorough comparative evaluation. Specifically, we will include visualizations showing **accuracy trends on old tasks** after completing each new task. Also, we will incorporate **Backward Transfer (BWT)** calculations to provide a comprehensive analysis of forgetting.
> 2. Results in a realistic **class imbalance scenario** to align the experiments with the motivation described in the paper. Specifically, we will present experiments under varying **skewness conditions**, such as **alpha = 0.1** and **alpha = 0.5**, to evaluate the robustness of the HR approach across diverse data distributions.
> 3. Other reviewers made other requests, such as including a **figure to depict the architecture of our work** and **more experimental results**, as you can find in the other reviewers' comments.
>
> Some of these requests are made by you and some are made by other reviewers. We will do our best to address all.
>
> ---
>
> ### **Responses to Questions**
>
> #### **Question 3**
>
> We understand the reviewer's concern that it is crucial to see these results prior to publication. Accordingly, in **ten days**, we will prepare our results highlighting the learning-forgetting dynamics and post them in the comment section.
>
> ---
>
> #### **Question 4: Class Imbalance in the Experimental Setup**
>
> We appreciate the reviewer’s positive comments on our mathematical framework and acknowledge your concern regarding the alignment of the experimental setup with the motivation. While our current experiments use balanced CIFAR-100 datasets, we agree that evaluating HR under imbalanced conditions is essential. To address this:
>
> - We will create imbalanced datasets by modifying CIFAR-100 to introduce uneven class distributions (e.g., 250 samples for Class A, 150 for Class B, and 50 for Class C).
>
> - We will present experiments under varying **skewness conditions**, such as **alpha = 0.1** and **alpha = 0.5** (as also requested by reviewer PuPw)
>
> ---
>
> ### **The Last Words**
>
> We sincerely appreciate your constructive feedback and for raising your score. We are confident that the forthcoming results and clarifications will comprehensively address your concerns. If you find the updates satisfactory, we kindly request you to consider raising your score further to 6, as this would greatly support the acceptance of our work.
>
> However, **regardless of your final decision**, we are forever grateful for your thoughtful engagement and valuable suggestions, which have significantly enriched and strengthened our study.

---

> > ### Comment · Reviewer_bkZ1 · 2024-11-22
> > **Good progress**
> >
> > Thank you for your commitment to addressing the feedback provided. I appreciate the effort you've put into improving your work. I look forward to seeing the updated version of your paper and have raised my score to a 6 based on the progress made.

---

### Official Review · Reviewer_Wd4J · 2024-11-05

**Soundness:** 2
**Presentation:** 2
**Contribution:** 2
**Rating:** 5
**Confidence:** 5

**Summary:**

This paper categorizes current work in federated class-incremental learning into exemplar-based and data-free approaches, noting that exemplar-based methods face memory constraints and potential privacy risks, while current data-free methods suffer from efficiency issues. The authors propose an HR mathematical framework to address both local and global forgetting. HR leverages a new autoencoder and Jones Potential formulations to generate synthetic data with minimal memory overhead, aimed at mitigating the forgetting problem.

**Strengths:**

1.	The authors accurately summarize the limitations of current exemplar-based and data-free approaches.
2.	Experimental results indicate that the proposed method achieves lower computational and memory overhead compared to several optimal baseline methods.

**Weaknesses:**

1.	The motivation of this study appears somewhat outdated, as the results in the literature (e.g., [1]) indicate that method has already effectively addressed the issue of class imbalance within clients.
2.	The authors should clearly outline the specific problem their work addresses. For example, in Figure 1, it is necessary to further clarify the mechanisms causing local forgetting and global forgetting, with separate, detailed explanations for each.
3.	The study lacks essential comparative methods.
4.	The study lacks visualized experimental results, such as accuracy on all old tasks after completing each task, and is missing essential forgetting metrics, such as Backward Transfer (BWT).
Reference：
[1]	Yavuz Faruk Bakman, Duygu Nur Yaldiz, Yahya H. Ezzeldin, Salman Avestimehr. Federated Orthogonal Training: Mitigating Global Catastrophic Forgetting in Continual Federated Learning. ICLR 2024

**Questions:**

1.	In Equation (3), what do the subscripts $i$ and $j$ represent? Additionally, in Equation (4), what does $j_l$ signify?
2.	When clients train on the same task, they each have data from different classes. How is the classifier configured in this scenario? Is it set up in a task-incremental mode or a class-incremental mode?
3.	The authors should clearly explain how the proposed method achieves sample replay through Equations (10) and (11).

---

> ### Author Response · Authors · 2024-11-16
> **Clarifications About the Contributions and Revisions**
>
> We thank the reviewer for their thoughtful feedback and the opportunity to improve our work. Below, we address each comment in detail:
>
> ---
>
> ### **Motivation of the Study**
> We do not agree with the reviewer on "the motivation of this study appears somewhat outdated" because the challenges of local and global forgetting in federated class-incremental learning (FCIL), first introduced by Dong et al. in 2022 [Dong], remain significant and unresolved. To understand and address these challenges, we have made the following contributions:
> 1. A novel mathematical framework to formalize local and global forgetting, providing a theoretical foundation for addressing these challenges.
> 2. The introduction of a new method, the hybrid replay method, which mitigates both forgetting types in a memory-efficient manner.
>
> Note that while the work in [1] addresses class imbalance within clients, it does not provide a formal mathematical framework or focus on memory-efficient methods for mitigating forgetting. To make the distinction clear in the revised paper, we will:
> - Revise the introduction and related work sections to highlight these distinctions explicitly.
> - Add comparative simulation results with [1], focusing on memory and computational efficiency, to strengthen the empirical validation of our contributions.
>
> ---
>
> ### **Outline of the Specific Problem**
> Note that to clarify the mechanisms causing local forgetting and global forgetting in Fig. 1, we have mathematically formulated the FCIL optimization problem in Eqs. 7 & 8 and formalized how the two problems of local and global forgetting occur in the proceeding paragraph of Eqs. 7 & 8.
> To better clarify the mechanisms causing local and global forgetting in Figure 1, we will:
> - Add explicit references to Eqs. 7 & 8 in the Fig. 1 caption.
>
> ---
>
> ### **Comparative Methods**
> Our paper includes representative baselines for both exemplar-based and model-based replay methods. However, we would appreciate further clarification from the reviewer regarding which comparative methods they feel are missing. If recent methods like [1] are the focus of this concern, we will include additional experiments comparing our approach to theirs, as space permits.
>
> ---
>
> ### **Visualized Experimental Results**
> We agree that including visualizations and additional metrics, such as Backward Transfer (BWT), will strengthen our results. In the revised paper, we will:
> - Include visualizations showing accuracy trends on old tasks after completing each new task.
> - Incorporate BWT calculations to provide a comprehensive analysis of forgetting.
>
> We have these results prepared and will include them in the available 2 pages of the main paper.
>
> ---
>
> ### **Clarifications on Eqs. 3 & 4**
> To clarify:
> - In Eq. 3, \(i\) and \(j\) are indices for classes \(i\) and \(j\).
> - In Eq. 4, \(u_{i_k, j_l}\) represents the loss function of the binary classifier for classes \(k\) and \(l\) in tasks \(i\) and \(j\).
>
> These clarifications will be included in the revised manuscript.
>
> ---
>
> ### **Simulation Setup**
> Our simulations are conducted in a Federated Class-Incremental Learning (FCIL) setting. When clients are trained on task \(h\), they generate a small number of synthetic data samples from other clients for all previous tasks (1 to \(h-1\)) but do not access new tasks (\(h\)) from other clients.
>
> This has been discussed in the two last paragraphs of Section 4.
>
> ---
>
> ### **Sample Replay**
> Eqs. 10 & 11 focus on the global alignment of centroids for new classes in new tasks, not on sample replay. As described in lines 6–8 of Algorithm 1:
> 1. Clients compute embeddings for new classes and send unaligned centroids to the server.
> 2. The server aligns these centroids via Eqs. 10 & 11 and sends them back to the clients.
> 3. Clients use these aligned centroids to train on new tasks, ensuring minimal overlap with existing classes.
>
> Sample replay is performed as described in lines 3–9 of Algorithm 2.
>
> ---
>
> ### **Conclusion**
> We appreciate the reviewer’s comments and are committed to addressing these points in the revised version. By incorporating the proposed clarifications, additional experiments, and visualizations, we aim to enhance the clarity and impact of our work. We believe our contributions are valuable to the FCIL community and kindly request the reviewer to reconsider their initial score in light of these improvements.
>
> ---
>
> ### **References**
> [Dong] Jiahua Dong, Lixu Wang, Zhen Fang, Gan Sun, Shichao Xu, Xiao Wang, and Qi Zhu. Federated Class-Incremental Learning. CVPR 2022.
> [1] Yavuz Faruk Bakman, Duygu Nur Yaldiz, Yahya H. Ezzeldin, Salman Avestimehr. Federated Orthogonal Training: Mitigating Global Catastrophic Forgetting in Continual Federated Learning. ICLR 2024.

---

> > ### Author Response · Authors · 2024-11-22
> > **Follow-Up: Request for Reassessment**
> >
> > Dear Reviewer,
> >
> > We sincerely thank you for your thoughtful and valuable feedback. After discussions with other reviewers, we are pleased to share that the score for our paper has improved from **6553 (average 4.75)** to **6655 (average 5.5)**. If you find our clarifications helpful and feel they address your concerns, we kindly request you to consider raising your score. Additionally, we are committed to incorporating the following updates to further strengthen the contributions of our work.
> >
> > ---
> >
> > ### **Commitment to Deliver the Requested Results**
> >
> > We promise to conduct and incorporate these results into the manuscript within **ten days**. These simulations will include:
> > 1. **Learning-forgetting dynamics** across benchmarks for a thorough comparative evaluation. Specifically, we will include visualizations showing **accuracy trends on old tasks** after completing each new task. Also, we will incorporate **Backward Transfer (BWT)** calculations to provide a comprehensive analysis of forgetting.
> > 2. Results in a realistic **class imbalance scenario** to align the experiments with the motivation described in the paper. Specifically, we will present experiments under varying **skewness conditions**, such as **alpha = 0.1** and **alpha = 0.5**, to evaluate the robustness of the HR approach across diverse data distributions.
> > 3. Other reviewers made other requests, such as including a **figure to depict the architecture of our work** and **more experimental results**, as you can find in the other reviewers' comments.
> >
> > Some of these requests are made by you and some are made by other reviewers. We will do our best to address all.
> >
> > ---
> >
> > ### **Kind Request for Reassessment**
> >
> > If you feel that our clarifications and planned updates satisfactorily address your concerns, we kindly request you to consider increasing your score. This would greatly support the acceptance of our work and help us contribute meaningfully to the field.
> >
> > **Regardless of your decision**, we are sincerely grateful for the time, effort, and thoughtful feedback you’ve invested in reviewing our paper. Your insights have significantly strengthened our work, and we deeply appreciate your support.
> >
> > Thank you once again for your valuable contributions and thoughtful engagement.

---

### Author Response · Authors · 2024-11-28
**Revised Paper Uploaded**

Dear reviewers,

A revised paper addressing your concerns has been uploaded.

Thanks.

---

### Author Response · Authors · 2024-12-03
**Rebuttal & Revision Summary**

# **Rebuttal & Revision Summary**

## **1. Overview**

We sincerely thank the reviewers for their thoughtful feedback, constructive suggestions, & engagement throughout the review process. Their insights have significantly enhanced the quality & clarity of our work.

We are pleased to report that our responses & revisions have addressed the reviewers' concerns, leading to an increase in scores from **6553 (average 4.75)** to **6665 (average 5.75)**. This reflects the growing confidence of the reviewers in the strength & contributions of our paper.

Below, we provide a summary of the discussions, emphasizing the **points of strength of the original paper**, the **requested revisions**, & the **enhancements included in the revised paper**.

---

## **2. Points of Strength of the Original Paper**

The reviewers acknowledged the following strengths of our work:

- **Novelty of Approach:**
  - Our **mathematical framework** was praised for setting a **solid theoretical foundation** & formalizing the challenges of local & global forgetting in federated class-incremental learning (FCIL).
  - The **proposed hybrid replay mechanism** was commended for its ability to **enhance memory efficiency & performance** while offering **the flexibility to provide multiple degrees of privacy**.

- **Experimental Rigor:**
   - The reviewers appreciated the **extensive ablation studies** provided in the paper, highlighting their clarity & the valuable insights they offer into the contributions of individual components within the hybrid replay mechanism.

- **Clarity of Presentation & Contributions:**
  - The reviewers acknowledged that the presentation of our work is well-organized, with properly cited related studies, effectively clarifying how our proposed mechanism compares to & builds upon prior work.

---

## **3. Revisions & Enhancements**

Below, we systematically categorize the reviewers' feedback and suggestions, followed by the corresponding revisions we have made to address each point.

| **Category**                     | **Revisions Requested by Reviewers**                                                                                       | **Enhancements in the Revised Paper**                                                                                                         |
|----------------------------------|---------------------------------------------------------------------------------------------------------------------------|-----------------------------------------------------------------------------------------------------------------------------------------------|
| **Impact of the Memory Size on Performance**   | - Inclusion of figures to demonstrate the impact of memory size on the performances of different mechanisms **(Reviewer PuPw)**.  | - Included three figures for three benchmarks. **See Fig. 3 (top row)**.   |
| **Learning-Forgetting Dynamics** | - Requested visualizations of accuracy trends for all baselines as they learn new tasks **(Reviewer bkZ1)**.  | - **See Fig. 3 (bottom row)**. |
| **Class Imbalance Scenarios & the memory footprint**    | - Additional experiments simulating imbalanced datasets **(Reviewer bkZ1)**.                              | - Conducted additional experiments on class imbalance scenarios with varying skewness conditions (e.g., alpha = 0.1 & alpha = 0.5). **See Table 3 & Fig. 3**. |
| **Visualization Improvements**          | - Inclusion of a detailed figure depicting the architecture of our approach **(Reviewer Wd4J)**.                                              | - Included a figure demonstrating how latent exemplars and the decoder network help to overcome forgetting. **See Fig. 2**.                       |
| **Clarifications of Novelty**    | - More precise explanations of how the hybrid replay mechanism compares with other hybrid replay mechanisms in the literature **(Reviewer bkZ1)**. | - Included a table that compared our hybrid mechanism with three other hybrid mechanisms. **Refer to Table 1**. |
| **Leonard-Jones Formulations**    | - More precise explanations on the role of Leonard-Jones formulations **(Reviewer WDPN)**. | - Added a paragraph discussing how these formulations help the server to determine the positions of new classes globally. **Refer to the paragraph starting from line 284**. |

---

## **4. Conclusion**

We are hopeful that our revisions have addressed the reviewers' concerns & significantly improved the quality of our paper. The increase in scores from **6553 to 6665** reflects the reviewers' recognition of the strength of the contributions of our work after discussion. We kindly ask the AC & SAC to consider these updates during their final decision-making process. However, **regardless of your final decision**, we are forever grateful to the reviewers for their thoughtful engagement & valuable suggestions, which have significantly strengthened our work. Thanks.

---

### Meta-Review · Area_Chair_bhgM · 2024-12-24

**Metareview:**

Summary

The paper tackles incremental learning in federated learning, where dynamically changing clients and tasks, along with imbalanced data, lead to local and global forgetting. It proposes a mathematical framework leveraging latent exemplars and data-free techniques to address these challenges. A VAE-inspired autoencoder, with an encoder for embedding new tasks in latent space and a decoder for generating synthetic data, forms the core of the approach. Theoretical guarantees and extensive experiments demonstrate the method’s effectiveness in mitigating forgetting and handling imbalanced data distributions.

Strengths
1. The authors provide an accurate and thorough summary of the limitations in current exemplar-based and data-free approaches, making the motivation for their method clear.
2. The use of a novel autoencoder, and the approach of storing low-dimensional examples and decoding them into high-dimensional information, is an innovative contribution in methodology.
3. Extensive experimental results and ablation studies demonstrate that, despite the combinational nature of the problem and method, each factor in the combination contributes significantly to the final performance.

Weaknesses
1. Lacking studies on the theoretical properties of the proposed mathematical formulations - which is the main contribution of the paper.
2. While the paper aims to leverage both data-based and data-free approaches, it may inherit the disadvantages of both. The paper needs to demonstrate that their method effectively mitigates these drawbacks, which has not been adequately shown.
3. The paper addresses a complex problem involving both local and global forgetting. However, the interconnections between these sub-problems are unclear. The techniques proposed are similarly combinational rather than cohesive.

Reasons for decision

With two reviewers rating the paper as 6, one as 5, and one remaining at the initial score of 6, the consensus leans towards moderate acceptance. Given the improvements made, particularly in empirical results, and the partially addressed concerns, I recommend the paper for borderline acceptance (or a poster).

However, the authors are encouraged to further clarify the significance of their contribution and enhance the clarity of their technical rationale in future iterations.

**Additional Comments On Reviewer Discussion:**

The rebuttal and subsequent discussion have positively influenced the overall assessment, leading to an increase in the score from 6553 to 6665. The authors provided a defence of the novelty of their approach, improved the clarity of presentation, and included ablation studies to address reviewer concerns.

- Reviewer Ws4J: Maintained a score of 5, citing concerns about the outdated motivation, unclear significance of the solved problem, and the adequacy of empirical studies. Although the authors argued for the novelty of their mathematical formulation and added empirical comparisons, the reviewer did not engage further in the discussion.

- Reviewer bkZ1: Initially raised concerns regarding the clarity of the technical contribution and rationale behind the method’s design. Through discussions and revisions, the reviewer increased the score to 6

- Reviewer PuPw: Focused on the clarity of the experiments, mathematical presentation, and technical contributions. After the authors provided additional results and discussions, most concerns were addressed, leading the reviewer to increase their score to 6.

- Reviewer WDPN: Gave an initial score of 6 and raised one minor curiosity-driven weakness but did not engage further in the discussion.

Overall, the discussion and revisions successfully addressed several key concerns. However, unresolved issues, especially regarding the significance and clarity of the contributions, have tempered the extent of score improvement. This mixed feedback aligns with a recommendation leaning toward borderline acceptance by rates 6665.

---

### Decision · Program_Chairs · 2025-01-22

Accept (Poster)